# Effect of Microstructure on the Mechanical Response of Hydrogen-Charged Pure Iron

**Boris Yanachkov [1,*], Lyudmil Lyutov [1,2], Ivaylo Katzarov [1], Ludmil Drenchev [1] and Krasimir Kolev [1,2]**

[1] Institute of Metal Science, Equipment, and Technologies with Center of Hydro- and Aerodynamics "Acad. A. Balevski" at the Bulgarian Academy of Sciences, 67 "Shipchenski prohod" str, 1574 Sofia, Bulgaria
[2] Faculty of Chemistry and Pharmacy, Sofia University "St. Kliment Ohridski", 1 James Boucher bul, 1164 Sofia, Bulgaria
* Correspondence: byanachkov@ims.bas.bg

**Abstract:** In this paper, we investigate how two different microstructures in pure iron affect the dislocation mobility in hydrogen-charged and non-charged samples by conducting stress-relaxation tests. The effective activation volume of the pure iron for both types of microstructures (cold-rolled and annealed samples) has been determined for both H-charged and uncharged material. Information about the dislocation structures formed during stress relaxation is provided by conducting TEM analysis. We employ a self-consistent kinetic Monte-Carlo (SCkMC) model of the ½ [111] screw dislocation in Fe to investigate how hydrogen affects the mobility and behavior of the dominant mobile dislocation in Fe at different stresses and H concentrations. The results from our simulations show the following: (i) at low stresses the deviation from the primary slip plane in the presence of H is lower than the deviation in the uncharged Fe. The deviation angle decreases with increasing H concentration; (ii) at higher shear stresses, the higher probability for kink-pair formation in the secondary (110) planes in the presence of H, leads to an enhanced deviation from the primary slip plane, which increases with increasing H concentration. We use the results of stress-relaxation tests and SCkMC simulations to propose an explanation for the formation of dislocation cell structures in pure and hydrogen charged Fe in the cold-rolled and annealed samples.

**Keywords:** hydrogen; dislocations; stress- relaxation tests; iron

## 1. Introduction

Hydrogen embrittlement has a significant role in the failure of individual engineering components and structures [1–3]. Studying this phenomenon and possibly predicting its occurrence is very important when working with iron and iron-based alloys, as they are the most widely used structural materials. When hydrogen is dissolved in steels even in a few atomic parts per million (appm), it causes a significant loss of fracture toughness. One possible explanation for this phenomenon is given by the hydrogen enhanced local plasticity (HELP) hypothesis [4]. At the atomic level, the reason for the plastic deformation of metals and alloys is the presence of dislocations. According to the HELP theory, in the presence of hydrogen, dislocations increase their mobility and speed [5–7]. This results in the formation of different dislocation structures in the individual metal grains in charged and uncharged samples. The result is a significant change in the overall mechanical properties. One macroscopic parameter that can describe the dislocation mobility is the effective activation volume. The activation volume $V_{eff}$ is the number of atoms that must be thermally activated to move dislocations over localized obstacles in their slip planes. This parameter cannot be directly measured. The effective activation volume for the studied material can be determined experimentally by carrying out a tensile stress-relaxation test [8]. This test is performed by loading a specimen beyond

the yield strength and holding it at a constant strain and recording the decrease in stress over a fixed period of time.

In order to observe the effect of hydrogen on dislocation behavior, the samples were electrochemically charged with hydrogen. All tests were performed on pure, initially cold-rolled iron. The same material has been used by other authors studying dislocation mobility in pure and H-charged Fe [9]. The authors who have conducted stress-relaxation tests to study the effect of H on $V_{eff}$ in Fe have not investigated the effects arising from the initial microstructure in pure Fe [5,9,10]. The aim of the present paper is to investigate how the initial microstructure affects dislocation mobility and formation of dislocation structures in pure and H-charged Fe. In the present work, all relaxation tests were performed with cold-rolled iron. One series of samples were tested as received and the others were heat-treated to obtain a different microstructure. We investigate how the different microstructure in pure iron affects the dislocation mobility in H-charged and uncharged samples. Additional information on the dislocation structure formed during the stress-relaxation tests is provided by using TEM observations. In this work, we use a self-consistent kinetic Monte-Carlo (SCkMC) model of the ½ [111] screw dislocation in Fe, developed in [9,11], to investigate how hydrogen affects the mobility and behavior of the dominant mobile dislocation in Fe at different stresses and H concentrations. By examining the experimental results and performing computer simulations, we propose an explanation for the formation of dislocation cell structures in pure, hydrogen-charged Fe in the cold-rolled and annealed state. The aim of the present paper is to study the effect of the initial microstructure of a sample, subjected to a stress-relaxation test, on the mobility of dislocations and the formation of dislocation structures in pure and H-charged Fe.

## 2. Materials and Methods

The starting material used in the study was pure iron in the form of 0.5 mm thick foil supplied by Goodfellow England. The purity of the sample is 99.9+% and the as-received condition is cold-rolled. The chemical composition of the sample was investigated with a spark optical emission spectrometer (OES) Q4 TASMAN Q101750−C 130, BRUKER AXS−Germany. Metallographic analysis of the samples was performed using a Leader series LM-308BD vertical metallurgical microscope with a digital camera. A furnace with an inert atmosphere (Ar- 99.999%) that can operate up to 950 °C was used for heat treatment of the samples. Electropolishing was carried out with a specially designed power supply with current limiting capabilities and operating voltage sufficient for the needs of the process (45–55 V). Mechanical tests were performed on a HA 250 ZWICK ROELL servo-hydraulic mechanical test system. The hydrogen charging of the sample is carried out using a specially designed electrochemical cell, that can operate during the mechanical test. This equipment is described in detail later in the article. After the tests, the samples were analyzed by TEM observation with a Tecnai G2 F20 (200 kV).

The authors who have investigated the effect of H on dislocation mobility in Fe by stress-relaxation tests [5,9,10] do not report how the initial microstructure of the samples affects $V_{eff}$. In order to account for the effect of the microstructure of the samples, we propose additional steps in the experimental approach designed to investigate the overall dislocation mobility and determination of $V_{eff}$.

### 2.1. Sample Characterization and Preparation

The material used to conduct the stress-relaxation tests was commercially available pure iron (Fe 99.9+%) in 0.5 mm thick plate form. The same material has been used by other authors to determine the effect of H on the effective activation volume in Fe [5,9,10]. Initially, the samples were subjected to spark OES analysis to confirm chemical composition. Small samples taken from the pure iron plates were embedded in resin and mechanically polished and then etched with a solution of 4% $HNO_3$ in ethanol for 15 s to observe their microstructure.

The sample preparation procedure consists of 3 steps: precision cutting of the samples, heat treatment (to obtain a reproducible structure), electrochemical polishing (to obtain a reproducible surface finish). To ensure that the samples had the same geometry, the pure iron plate (100 × 100 mm) was cut by wire electrical discharge machining (EDM). The dimensions of the sample are shown in Figure 1. This cutting method produces the least mechanical distortion and almost no thermal effects on the material near the cut.

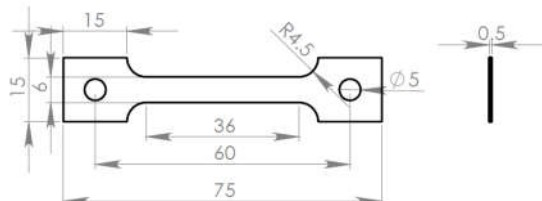

**Figure 1.** Dimensions of specimen for tensile tests.

In order to obtain a different microstructure, the samples were heat treated in an inert atmosphere furnace (Ar 99.999%). Several different heat treatment regimes were tested on samples of the same composition. The results were analyzed and the following optimal regime has been determined: 1.5 h temperature rise to 900 °C, 1 h hold time at 900 °C and 4 h temperature decrease from 900 °C to room temperature.

Different surface roughness is usually achieved during sample preparation. In terms of electrochemical hydrogen charging, this means that samples with different surface roughness (surface area) are likely to have different hydrogen contents and different mechanical responses. To prevent surface roughness from becoming another variable in the study, the samples were subjected to electrochemical polishing. This treatment guarantees good repeatability of the surface characteristics between samples. For this purpose, the iron strips were fixed in specially designed holders with electrically insulating PTFE masks. The test specimen holder assembly was immersed in the electropolishing solution and connected as the anode. The samples were electropolished in a solution containing 20:1 glacial acetic acid (p.a; Merck) to perchloric acid (70%, p.a; Merck). The procedure is carried out at 20 °C at an anode current density of 40 A/dm² for a period of 15 to 20 s.

### 2.2. Hydrogen Charging

It is well known that charging a sample with hydrogen and subsequent testing is accompanied by significant technical difficulties and quite a large uncertainty about the amount of hydrogen in the sample. This problem has been solved by constructing a specialized electrochemical cell for charging the sample with hydrogen simultaneously while conducting the mechanical tests. Thus, by controlling the operating parameters of the cell, it is possible to reach a good equilibrium and repeatable hydrogen concentrations. A schematic diagram of the electrochemical cell operated during the mechanical tests is shown in Figure 2. The test specimen 1 is placed in holders 2 connected to the mechanical testing machine. On both sides of the flat test specimen there are inert electrodes 3 with a surface area greater than that of the test specimen. Electrically insulating caps 4 made of PTFE are used to prevent current from flowing through the holders. The waterproof housing for the electrolyte 5 is made out of polypropylene. During operation, the test specimen is connected to the negative terminal of the DC power supply, and the inert electrodes to the positive. A special balancing resistor is connected to the positive electrodes. The resistance of this component is calculated to be within the cell resistance range. The purpose of the balancing resistor is to distribute the current evenly between the two anodes so that the sample is charged equally on both sides. This is measured by connecting a milliammeter in series with each anode. The value is adjusted so that both

milliammeters measure the same value. Tests are performed with a cathodic current density of 0.1 mA/cm².

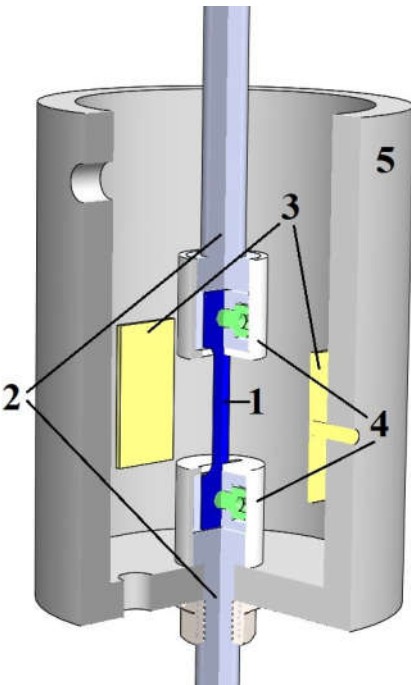

**Figure 2.** Schematic diagram of electrochemical cell for hydrogen charging during tensile test; (1) test specimen; (2) mechanical holders for the specimen; (3) positive electrodes; (4) electro insulating PTFE covers; (5) hermetic polypropylene enclosure for electrolyte.

The constant circulation of the electrolyte through external heat exchanger ensures the maintenance of stable operating conditions at 20 °C. The electrolyte used in charging is 0.5 mol/l sulfuric acid and 0.65 mol/l thiourea (promoter). The cell holders are connected to the machine by threads with locking nuts. A view of the cell placed in the mechanical testing machine is shown in the Figure 3.

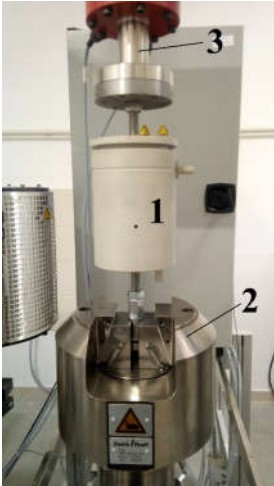

**Figure 3.** View of the electrochemical charging cell mounted in the mechanical testing machine: (1) cell; (2) moving jaw; (3) load cell connected to stationary beam.

The configuration, consisting of a universal testing machine equipped with an electrochemical cell, allows very accurate force measurement. This makes it possible to measure the elongation of the sample in a hydrogen-charged state and to determine qualitatively when the sample reaches maximum hydrogen saturation, under the specified conditions.

A dilatation experiment has been conducted to determine the time for electrochemical charging of the sample with hydrogen at a given cathodic current density (0.1 mA/cm²). The force exerted on the sample was adjusted to 0 N. A few seconds after the start of hydrogen charging procedure, a negative force is registered, which corresponds to the elongation of the sample. The testing machine is set to automatically compensate for the negative force by increasing the distance between the holders. The elongation of the sample with respect to time is recorded.

*2.3. Mechanical Tests*

To properly measure the effect of hydrogen on the mechanical behavior of pure iron, it is first necessary to determine the yield stress of the material from the corresponding stress-strain curve. This has been achieved by conducting a standard tensile test with the specimens. The yield stress is a criterion for the state of the material and its microstructure due to the strong relationship between the microstructure and the mechanical properties. The value of the yield stress is used in subsequent stress-relaxation tests, which are carried out by increasing the stress for the first cycle to a value that is above the yield point.

The stress relaxation test is carried out at room temperature by fixing the sample in the electrochemical cell without electrolyte and increasing the tensile stress for the first cycle to a value that is above the yield stress by 30 MPa. The same strain is then maintained for 20 s and the stress drop is recorded as a function of time. In the subsequent cycles, the specimen has been subjected to the same initial tensile stress applied at a fixed strain rate of 0.008 s⁻¹, followed by maintaining the strain again for 20 s. This cycle is repeated 10 times for each specimen. Both types of samples were subjected to this test without hydrogen-charging. The second phase of the experiment was conducted with both types of samples charged with hydrogen. The samples were placed in the holders of the electrochemical cell filled with electrolyte. Formation of small bubbles of hydrogen was observed on the surface of the samples when charging is initiated. To ensure stable temperature conditions during charging, the electrolyte was constantly circulated in an external heat exchanger at 20 °C. The samples were charged for a period of 1 h.

These tests were conducted by loading a sample beyond the yield strength and holding it at a constant strain and recording the decrease in stress over a fixed period of time. This technique measures apparent activation volume $V_a$ that is related to $V_{eff}$ by Equation (1):

$$V_a = V_{eff} + V_h \tag{1}$$

$V_{eff}$ is accessible by use of the stress-relaxation technique only if the strain-hardening term $V_h$ is known. $V_h$ can be determined from the successive stress drops of the relaxation series with constant duration [8] by Equation (2):

$$(n - 1)V_h = \frac{kT}{\overline{\Delta\tau}} ln \left[ \frac{exp(-\Delta\tau_n/\mu) - 1}{exp(-\Delta\tau_1/\mu) - 1} \right] \tag{2}$$

where $\Delta\tau$ is the stress drop, the subscript $n$ is the relaxation number of the series, $\overline{\Delta\tau}$ is the mean stress drop value and $\mu = kT/V_a$. The stress decrease can be expressed by the logarithmic relation Equation (3)

$$\Delta\tau_i = -\mu ln \left( 1 + \frac{t}{c} \right) \tag{3}$$

where $c = \mu/E\dot{\gamma}_i$ is an integration constant which can be determined at the onset of the relaxation. E is the elastic modulus of the sample-machine assembly and $\dot{\gamma}_i$ is the plastic

strain rate at the onset of relaxation. Within the relaxation method, $V_a$ can be measured on the first relaxation, while $V_h$ can be determined from the stress decrease at the end of each subsequent relaxation [8].

### 3. Results

Spark OES analysis confirmed the compositional data provided with special attention to some elements in the iron. The results of the analysis are shown in Table 1.

**Table 1.** Chemical composition of starting material obtained with spark OES analysis.

| Element | C | S | P | Si | Mn | Cr | Ni | Cu | Fe |
|---|---|---|---|---|---|---|---|---|---|
| Average mass % | 0.0081 | 0.0053 | 0.0085 | 0.0133 | 0.0190 | 0.0300 | 0.0320 | 0.0155 | Bal. |

After the samples were prepared and etched, they were observed with an optical microscope. The observation results are shown in Figure 4.

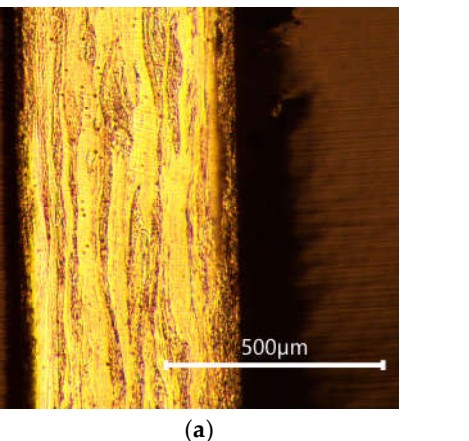 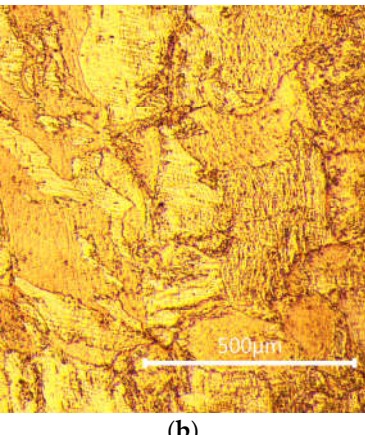

(**a**)      (**b**)

**Figure 4.** Optical micrograph showing the microstructure of the starting material: direction of rolling (**a**) perpendicular; (**b**) in parallel with the viewing plane.

The cold-rolled samples were found to have highly elongated crystals oriented in the direction of rolling. A pre-processing procedure is required to obtain reproducible results. As described above, the samples were subjected to heat treatment and then their microstructure was investigated. The microstructure before and after the heat treatment is shown in Figure 4 and Figure 5.

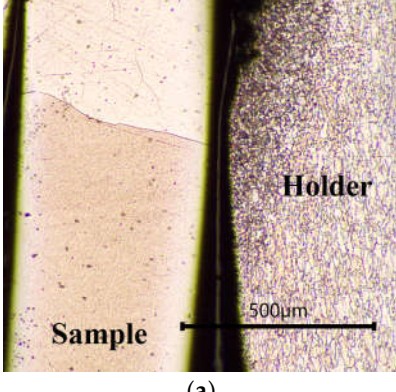 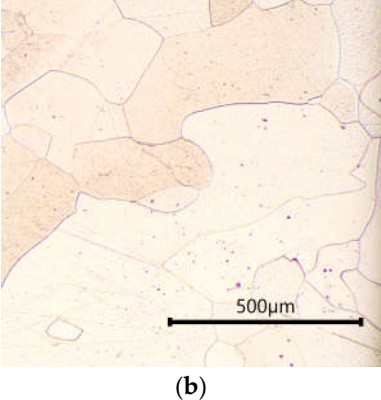

(**a**)      (**b**)

**Figure 5.** Optical micrograph showing the microstructure of the thermally treated material: direction of rolling (**a**) perpendicular; (**b**) in parallel with the viewing plane.

From a highly elongated form with obvious orientation (Figure 4), the grains recrystallize to become large and randomly oriented. This means that after recrystallization there is a significant difference in the amount of grain boundaries in the volume of the material. The internal stresses of the material are minimal and the structures generated by previous treatments are completely erased. Dilatometric measurements of sample elongation during hydrogen charging were performed for the two samples with different microstructures—cold-rolled and heat treated. The actively charged zone of both samples was 36 mm. The results are shown in Figure 6.

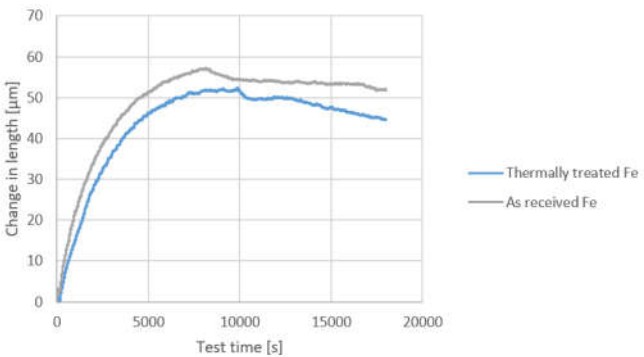

**Figure 6.** Dilatometric measurement of hydrogen-charging.

It can be observed that after a certain period (about 2 h) the samples achieve maximum elongation. This behavior can be attributed to reaching a state of full saturation at the applied current density and temperature. In order to determine the value of the tensile stress to which the samples should be subjected in the subsequent stress- relaxation tests, the yield strength of the material has been determined from the corresponding stress-strain curves Figure 7. It can be seen that due to the difference in microstructure, the two types of material have completely different mechanical properties.

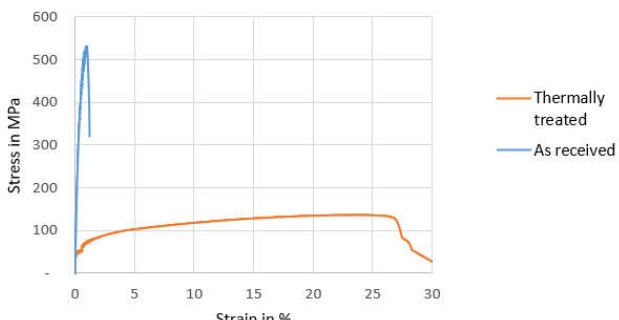

**Figure 7.** Stress- strain curve for pure Fe.

The experimentally determined relaxation curves of the cold-rolled samples with and without hydrogen are shown in Figure 8.

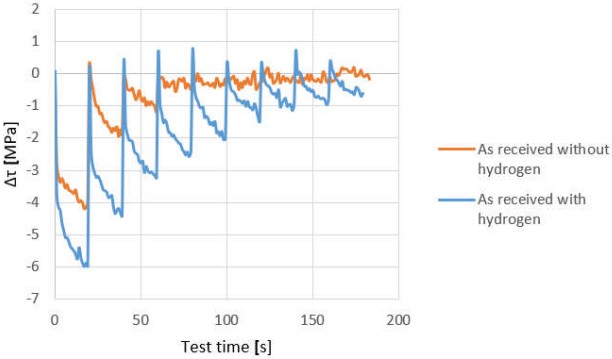

**Figure 8.** Stress- relaxation test of Fe in the as received state.

It should be noted that the value of the tensile stress is selected based on the measured yield stress. The samples were loaded to a stress which is above the yield limit of the material—151 MPa. The test results show that the uncharged sample rapidly loses its stress-relaxation ability after the 4th cycle. The hydrogen charged sample behaves differently. It can be seen from Figure 8 that it has ability to relax the stress in all the cycles it is subjected to.

Stress-relaxation tests were carried out with heat-treated samples in hydrogen charged and uncharged state. This type of samples has been loaded to 22 MPa, which is also above the yield point of this material. The experimentally determined stress curves are shown in Figure 9. From the experimental results, it can be seen that unlike the cold-rolled samples, the heat-treated ones have higher relaxation ability and retain their relaxation ability during all test cycles. It should also be noted that, in contrast to cold-rolled case Figure 8, higher stress decrease during relaxation cycles is observed in the samples without hydrogen (Figure 9). These observations are confirmed by the determined effective activation volumes.

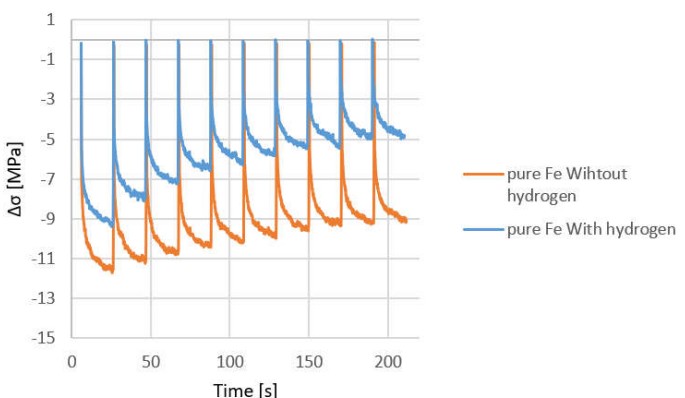

**Figure 9.** Stress-relaxation test of Fe in the thermally treated state.

Our measurements of $V_{eff}$ of the cold-rolled samples show that, of the hydrogen charged specimen, $138 \cdot b^3$ (b is the Burgers vector of ½[111] dislocation), is smaller than that of the uncharged samples, $146 \cdot b^3$. On the contrary, the effective activation volume of the hydrogen-charged sample annealed at 900 °C, $124 \cdot b^3$, is higher than $V_{eff}$ of the hydrogen-free sample, $110 \cdot b^3$.

### 4. Discussion

In [6,9], a self-consistent kinetic Monte-Carlo (SCkMC) model describing movement of ½ [111] screw dislocation in Fe-H system via Peierls (kink-pair) mechanism was developed. The SCkMC model provides a realistic description of the dynamics of a dislocation line over long time scales. The features of the SCkMC approach are as follows: (i) nucleation of kink-pairs followed by migration of kinks in opposite directions along the dislocation line; (ii) kink-pair formation energy depends on the concentration and mobility of H atoms trapped in the dislocation core; (iii) kinks require thermal activation to overcome the energy barrier associated with hydrogen atoms trapped just behind the core. Kink pairs can nucleate in any of the three intersecting (110) glide planes with the nucleation rates dependent on the magnitude and the direction of the maximum resolved shear stress (MRSS). Within the SCkMC model the ½ [111], dislocation moves under the action of applied shear stress through kink-pair nucleation, migration, collision and recombination. Details of the model of dislocation motion adopted by the authors are presented in [6,9].

It has been commonly accepted that a requirement for the formation of dislocation cells is that dislocations have sufficient mobility out of their slip planes [12,13]. Therefore, whether cells form or not depends on factors that determine the ease with which dislocations cross-slip or climb.

We use a SCkMC simulations to study the mobility of an individual ½ [111] screw dislocation in hydrogen-charged and uncharged $\alpha$-Fe. Simulations show that the behavior of the dislocation depends significantly on the applied stress and hydrogen content in the bulk. The angle describing the deviation of the dislocation from the primary slip plane at an applied stress $\tau$ = 30 MPa and 200 MPa as a function of the average distance covered by the dislocation is shown in Figure 10 in both hydrogen-free and hydrogen-charged conditions. The results show that, at lower τ with a hydrogen-free condition, the deviation angle remains constant with increasing distance. However, for the hydrogen content of 30 appm, the deviation angle rapidly reduces after the distance increases to 3 nm, Figure 10a. As the applied shear stress increases, we observe the opposite picture. In uncharged bcc-Fe the ½[111], screw dislocation practically moves in the primary (110) slip plane. At hydrogen content of 30 appm, the deviation angle rapidly increases to 26° (Figure 10b). These results were obtained for applied shear stress in which the MRSS plane bisects the primary glide plane at an angle of 15°. This orientation of the MRSS plane approximately corresponds to the average dislocation mobility resulting from different shear stress orientations [6].

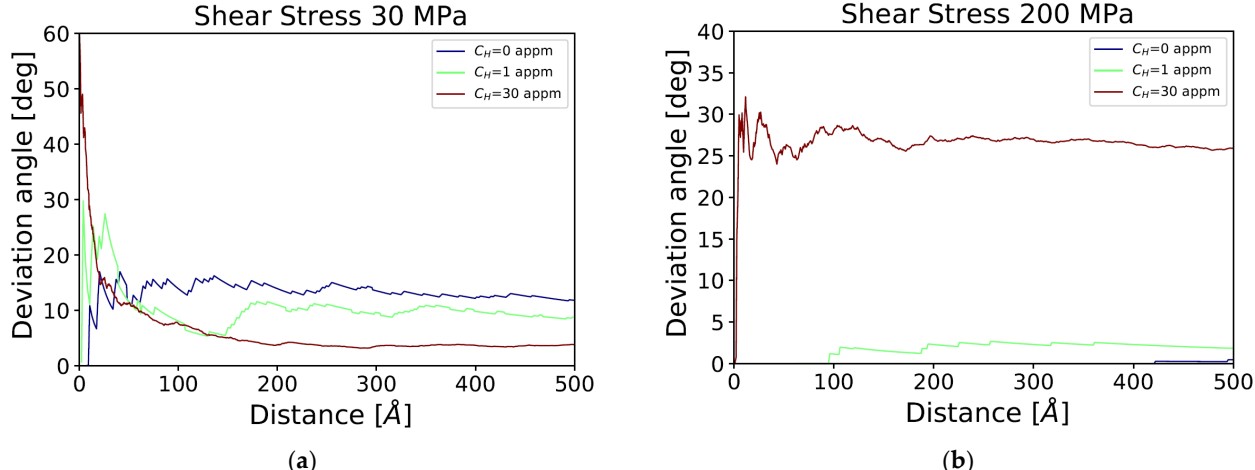

**Figure 10.** Deviation of the dislocation from the primary slip plane at an applied stress (**a**) τ = 30 MPa and (**b**) 200 MPa as a function of the average distance.

The kink-pair formation energy $E_{kp}$ depends on both applied shear stress and hydrogen concentration. $E_{kp}$ is large at low stress and H concentrations, the largest being that of pure bcc-Fe with no applied stress. Since the secondary Peierls barrier in pure Fe is low, kink migration is not thermally activated and is controlled by the phonone drag. Thus, due to the fast kink migration speed the kink reaches the dislocation line end with a typical length of 1000.b (b is the Burgers vector of ½ [111] dislocation) before the next kink pair is nucleated [9]. With $E_{kp}$ depending on the corresponding RSS, the kink pair formation in the secondary (110) glide planes leads to cross-slip of the dislocation. After formation of a kink-pair on the primary slip plane, dislocation cross-slip occurs again, resuming slip parallel to the primary plane. This zigzag motion of the dislocation leads on average to glide out of the primary glide plane. However, a hydrogen atom trapped in the dislocation core ahead of the kink induces an energy barrier [7], that requires thermal activation for the kink to proceed its migration along the dislocation line. The reduction of the average kink velocity due to hydrogen-trapping effect increases the probability for the collision of kinks propagating simultaneously in different planes and creation of jogs. The significant reduction of kink velocity along the secondary slip planes in charged Fe, resulting from the H trapped in the core and lower RSS, lead to a reduced dislocation glide out of the primary slip plane [9]. The results from our simulations indicate that, at low stresses, the misorientation of the screw dislocation decreases with hydrogen-charging condition, and the movement in the direction of the secondary glide plane is pretty well suppressed. According to our model prediction, at low shear stresses the deviation from the primary slip plane in the presence of H is lower than the deviation in the uncharged Fe. Hence, the probability for formation of a cell structure in hydrogen-free condition is expected to be higher than in hydrogen-charged iron. At higher shear stresses (above 150 MPa) SCkMC simulations [9] show that kink velocity is high even in the secondary glide planes, which, in combination with the higher probability for kink-pair formation in the secondary (110) planes, leads to enhanced deviation from the primary slip plane in the presence of H.

At higher shear stresses, TEM observations also show formation of dislocation cell structure in H-charged Fe [10]. Therefore, SCkMC simulations indicate that the ease with which ½ [111] screw dislocations cross-slip and glide out of the primary slip plane in hydrogen-charged bcc-Fe depends on the combination of the applied shear stress and H concentration.

The standard tensile tests for flat specimens reveal that the value of the yield point of the cold-rolled specimen is significantly higher than the yield point of the annealed sample. Since the stress-relaxation tests are performed by rising the stress for the first cycle to a value that is above the yield point, based on our SCkMC simulations, we might expect that the dominant mobile ½ [111] screw dislocations glide more easily out of the primary slip plane in hydrogen-charged bcc-Fe annealed samples and in H-free cold-rolled specimens. The results of SCkMC simulations are fully consistent with TEM observations showing formation of dislocation cell structure in hydrogen-charged Fe in the cold-rolled sample and in pure iron annealed at 900°C (Figure 11).

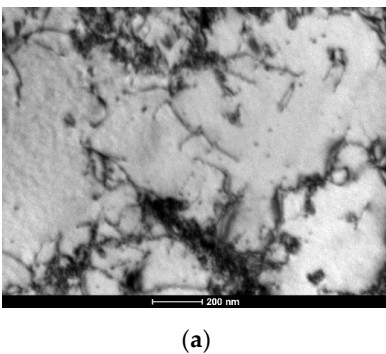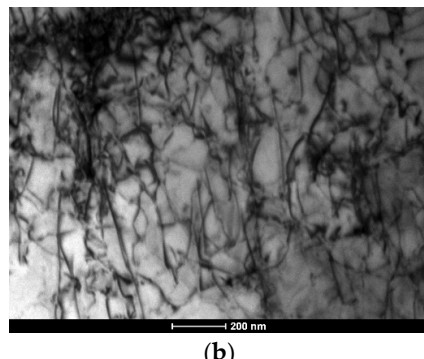

(**a**)　　　　　　　　　　　　　　　　(**b**)

**Figure 11.** TEM micrographs showing dislocation structure of Fe in annealed state (**a**) without hydrogen (**b**) hydrogen-charged.

Our measurements of the effective activation volumes in cold-rolled and annealed samples show that in both cases $V_{eff}$ is smaller when dislocations form cell structure (dislocations cluster into cell walls that separate relatively dislocation-free regions). The walls can be regarded as dense dislocation tangles. The driving force for formation of dislocation cell structure has been recognized as arising from a reduction in the total elastic energy of the dislocations because of the clustering. This results in the reduction of energy stored in the material and further plastic deformation can be undertaken [14].

The applied stress in combination with the internal stresses inside the tangles and cell walls, provide a driving force for a rearrangement of the dislocations into lower-energy configurations (such as dipoles and incipient subboundaries), which are also presumed to lower the local glide resistance. The microstructure at the end of the recovery processes discussed here would correspond to sharp cell walls on the same scale as the previous cell walls. Taking this into consideration, we could assume that the measured lower $V_{eff}$, when dislocations cluster into cell walls, is associated with further dislocation motion and the resultant evolution of dislocation structures which lower the local glide resistance.

## 5. Conclusions

In this paper we perform stress-relaxation tests to investigate dislocation behavior and mobility in hydrogen-charged and non-charged pure iron samples. Measurements of the effective activation volume for two types of microstructures show that dislocation mobility is higher in hydrogen-charged cold-rolled iron and in heat-treated hydrogen-free samples. TEM analysis shows formation of a dislocation cell structure in hydrogen-charged cold-rolled Fe and in heat-treated samples without hydrogen. SCkMC simulations of the mobility and behavior of an individual ½ [111] screw dislocation were carried out for various applied shear stresses and H concentrations. At low shear stress, corresponding to the relaxation test of heat-treated Fe, simulations show that the deviation of the dislocation from the primary slip plane in the presence of H is lower than the deviation in the uncharged Fe and decreases with increasing H concentration. At higher shear stresses, corresponding to the relaxation tests on cold-rolled Fe, screw dislocations glide out of the primary slip plane more easily in the presence of H. Since, a requirement for formation of dislocation cells is that dislocations have sufficient mobility out of their slip planes, in agreement with TEM observations, we conclude that the probability of formation of a dislocation cell structure is expected to be higher in pure Fe cold-rolled and hydrogen-charged annealed samples.

**Author Contributions:** Conceptualization, I.K. and L.D.; methodology, L.L.; investigation, B.Y. and K.K.; project administration, I.K. and L.D. All authors have read and agreed to the published version of the manuscript.

**Funding:** This research was funded by the Bulgarian National Science Fund (BNSF) under Programme grant KP-06-H27/19 and Project BG05M2OP001-1.001-0008, funded by the Operational Programme: Science and Education for Smart Growth, and co-financed by the European Union through the European Regional Development Fund.

**Data Availability Statement:** Not applicable.

**Acknowledgments:** The authors of this paper acknowledge the support from their colleagues from the Institute of Metallurgy and Materials Science of the Polish Academy of Sciences for the TEM observation of the specimens. The authors acknowledge the support from Mihail Kolev Boris Drenchev, Jordan Georgiev, Vania Diakova, the colleagues from the Faculty of Chemistry and Pharmacy at Sofia University "St. Kliment Ohridski"; Bulgaria Tony Spassov and Vesselina Rangelova and the colleagues from the Institute of Information and Communication Technologies at the Bulgarian Academy of Sciences. The authors are grateful for the support from the colleagues from the Institute of Mechanics at the Bulgarian Academy of Sciences.

**Conflicts of Interest:** The authors declare no conflict of interest. The funders had no role in the design of the study; in the collection, analyses, or interpretation of data; in the writing of the manuscript; or in the decision to publish the results.

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
