# Peer review of "Effect of Microstructure on the Mechanical Response of Hydrogen-Charged Pure Iron"

_metals, doi:10.3390/met12122160_

Round 1
Reviewer 1 Report
In this manuscript the authors studied the relationship between microstructures in pure iron and the dislocation mobility in hydrogen-charged and non-charged samples. The research work is rather systematical and the results may be beneficial to the researchers working in the relevant field. I suggest the manuscript may be accepted for publication in Metal. However, the following revisions are required.
(1) The Abstract needs to be rewritten. At current stage, it mainly contains the description of the experimental details without conclusive content. Also, the Conclusion part needs be shortened and simpler.
(2) The content between lines 78-80 should not be appeared in the Materials and Methods part. The aim of this work needs to be placed in the last paragraph of the Introduction part.
(3) I strongly suggest the authors to carefully check the format and writing.
Author Response
We express our gratitude to the referee for the careful reading of the paper and useful comments. The paper is modified with the referees comments taken into account. The corrections are given in Bold font. The modified version is attached here.
Please find below our response to the reviewers comments.

Reviewer 2 Report
REVIEW REPORT MDPI Metals (metals- 2084987)
Effect of microstructure on the mechanical response of
hydrogen-charged pure iron
Boris Yanachkov, Lyudmil Lyutov, Ivaylo Katzarov,Ludmil Drenchev, Krasimir Kolev
In this work , the authors present the results of the study of how two different microstructures in pure iron affect the dislocation behavior and mobility in hydrogen-charged and non-charged samples by conducting stress-relaxation tests. Information about the dislocation structures formed during the stress relaxation obtained by conducting TEM analysis. The authors used a self-consistent kinetic Monte-Carlo model of the ½[111] screw dislocation in Fe to investigate how hydrogen affects the mobility and behavior of the dominant mobile dislocation in Fe at different stresses and H concentrations.
The article is written in clear, understandable language.
The article contains new original results and matches the profile of the journal.
Essentially, there are several suggestions and wishes for the authors:
100-105 It is desirable to explain why the processing mode "1.5 hours temperature rise to 900°C, 1 hour hold time at 900°C and 4 hours temperature decrease from 900°C to room temperature" is selected.
-This is an area of polymorphic and phase transformations, what is the physicochemical meaning of this processing mode?
118, 138, 144,182 "Hydrogen charging: cathodic current density of 0.1 mA/cm2, stable operating conditions at 20°C, 0.5 mol/l sulfuric acid and 0.65 mol/l thiourea. The samples were charged for a period of 1 hour."
- Nice and useful cell for hydrogen charging. But how uniformly is hydrogen distributed in the sample during electrochemical charging? Does hydrogen accumulate in the near-surface region of micron-thick iron? It is interesting to compare the results of the authors with the results obtained for samples saturated by the Sieverts method. Electrochemical saturation does not guarantee a uniform distribution of hydrogen over a thickness of 0.5 mm, despite the high diffusion coefficient of hydrogen in iron (D(20°C)=1.5·10-5(cm2s-1 )
278-291
- It would be useful to add a figure explaining the model of dislocation motion adopted by the authors.
The article can be published without significant changes
Regards, Reviewer
04.12.2022
Author Response

(The authors gave the same response as above.)

Reviewer 3 Report
The paper offers a research, which is relevant to advancement of the understanding about the effects of hydrogen in metals, and the hydrogen embrittlement in particular. Accordingly, it is well suited to publication in the Metals Journal.
The paper presents a well-done research, which is also well organized and well written. Accordingly, I am delighted to recommend this work for publication.
Nevertheless, I have few doubts and minor suggestions. Attending them by the authors, I believe, could improve the manuscript.
I ask you kindly to clarify next issues of the paper technical content.
1) The dilatometric measures in Fig. 6 present the “change in length”. I wonder, what is the length, which change is measured, i.e., the unchanged length?
2) The Monte-Carlo simulations were performed, if I understood all well, at the lattice H concentration of 30 appm, which is 3e-05 at.H/at.Fe. Isn’t this too high to compare the theory with your experiments? Rough estimation of the lattice H concentration under the input H2 fugacity of 15 MPa basing on the commonly employed solubility data for H in Fe (see, e.g., the paper by Dadfarnia & Sofronis, Int. J. of Hydrogen Energy 2011, v. 36 p. 10141, doi 10.1016/j.ijhydene.2011.05.027) brings about 3e-07 at.H/at.Fe, that is, two orders of magnitude less. Then according to the Sieverts law, your concentration should correspond to the fugacity 15e+04 MPa. Does this correspond somehow to your charging system? Can it? Could you offer an estimation of the input hydrogen fugacity provided by your charging system?
Some points as regards the writing and manuscript formatting could be improved.
- The abbreviations (excepting quite commonly used) should be explained at the first use both in the text and in the abstract (this latter might be the only ever read passage of the paper). SCkMC lacks this in the abstract. OES, EDM, etc. lack explanation in the main text.
- Describing Fig. 2 in t he text, it would be better to number the picture elements in the order that they are explained, I mean first explain the item #1, then #2, etc., but not put #5 to the first explained item.
- I failed to understand the phrase in lines 164-165: “The yield stress is a criterion for the state of the material and its microstructure, due to the strong relationship between the microstructure and the mechanical properties.” Could you be more explicit, please?
- Delta-tau in eq. (2) has subscripts, but its explication in the text has not. Please, correct this.
- The multiplication dot (e.g., like 138.b3, etc. in lines 256-259) should be in the middle of the line, but not at the bottom, which is the place for decimal dot.
- The last section “Conclusions” provides merely a synopsis of what was done in the work, but really not the per se Conclusions from the work. Then, the section title should be adjusted to its present content or vice versa.
- If I’m not mistaken, the formatting of references does not follow the editorial requirements.
Author Response

(The authors gave the same response as above.)
